# Novel Host Response-Based Diagnostics to Differentiate the Etiology of Fever in Patients Presenting to the Emergency Department [note 1]

**DOI:** 10.3390/diagnostics13050953

**Published:** 2023-03-02

**Authors:** Johnny Atallah, Musie Ghebremichael, Kyle D. Timmer, Hailey M. Warren, Ella Mallinger, Ellen Wallace, Fiona R. Strouts, David H. Persing, Michael K. Mansour

**Affiliations:** 1Department of Medicine, Harvard Medical School, Boston, MA 02115, USA; 2Infectious Diseases Division, Massachusetts General Hospital, Boston, MA 02114, USA; 3Ragon Institute of MGH, MIT and Harvard, Cambridge, MA 02138, USA; 4DH Diagnostics LLC, Brea, CA 92821, USA

**Keywords:** host-response, diagnostics, gene signatures, fever, febrile syndromes

## Abstract

Fever is a common presentation to urgent-care services and is linked to multiple disease processes. To rapidly determine the etiology of fever, improved diagnostic modalities are necessary. This prospective study of 100 hospitalized febrile patients included both positive (FP) and negative (FN) subjects in terms of infection status and 22 healthy controls (HC). We evaluated the performance of a novel PCR-based assay measuring five host mRNA transcripts directly from whole blood to differentiate infectious versus non-infectious febrile syndromes as compared to traditional pathogen-based microbiology results. The FP and FN groups observed a robust network structure with a significant correlation between the five genes. There were statistically significant associations between positive infection status and four of the five genes: IRF-9 (OR = 1.750, 95% CI = 1.16–2.638), ITGAM (OR = 1.533, 95% CI = 1.047–2.244), PSTPIP2 (OR = 2.191, 95% CI = 1.293–3.711), and RUNX1 (OR = 1.974, 95% CI = 1.069–3.646). We developed a classifier model to classify study participants based on these five genes and other variables of interest to assess the discriminatory power of the genes. The classifier model correctly classified more than 80% of the participants into their respective groups, i.e., FP or FN. The GeneXpert prototype holds promise for guiding rapid clinical decision-making, reducing healthcare costs, and improving outcomes in undifferentiated febrile patients presenting for urgent evaluation.

## 1. Introduction

Fever is among the most common presentations to healthcare facilities and urgent care services [1]. Fever has been linked to multiple disease processes that include malignancy, autoimmune diseases, sterile inflammatory processes, and, most commonly, infectious etiologies [2]. Confirming the definitive etiology of fever in hospital-admitted adult patients remains a significant challenge for patient management and represents a large investment in hospital resource utilization [3]. When suspecting an infectious process, it is particularly challenging that clinical manifestations of microbial infection can range across a broad spectrum of nonspecific symptoms. Clinically, the detrimental impact of missed or delayed diagnoses can lead to antibiotic overuse, increased antimicrobial resistance, increased length of stay, and ultimately unfavorable health effects [4,5]. Economically, increased healthcare costs, prolonged hospitalization periods, and the excessive utilization of hospital resources directly result from poorly performing diagnostics [6]. The need to determine the etiology of fever in febrile patients at an accurate and rapid rate necessitates improved diagnostic modalities for better medical decision-making and management [7].

Pathogen-based diagnostic testing remains the current standard of care when dealing with suspected infections. However, as with any other diagnostic modality, pathogen-specific diagnostics have shortcomings. Depending on the pathogen-based diagnostic test chosen, these challenges include both low sensitivity or low specificity, especially for the commonly used blood cultures, often leaving the healthcare provider with a conundrum regarding the nature of the microbe detected, whether it is commensal or contaminant versus a true invasive pathogen [8,9].

Thus, as an adjunct to pathogen-specific diagnostics, host immunodiagnostics shows promise. These techniques involve assays such as reverse transcription-polymerase chain reaction (RT-PCR) to measure specific host gene expression signatures and transcripts, which can inform the host’s susceptibility and response to infection [10], allowing for the integration of multiple determinations into single predictive models for accurate diagnosis and disease prognosis.

Numerous gene signatures have been linked to the direct response of the immune system to different etiologies, including viral and bacterial infections. For example, the Interferon Regulatory Factor 9 (IRF-9) and Lymphocyte Antigen 6 Family Member E (LY6E) play an essential role in anti-viral immunity, including virus-mediated activation of interferons [11,12]. On the other hand, other genes, such as Integrin Subunit Alpha M (ITGAM), have been shown to regulate neutrophil migration and mediate the adhesion of neutrophils to pathogens leading to pathogen clearance in bacterial infections [13].

Other assays aimed at understanding the host response to different etiologies include flow cytometry. Flow cytometry can serve as an assay for stratification and differentiating the etiology of fever. Surface markers such as CD64 on neutrophils and CD169 on monocytes have been shown to serve as sensitive markers to differentiate bacterial infections from viral infections [14]

This prospective study aims to understand the transcriptional gene changes of circulating white blood cells in admitted patients with undifferentiated febrile syndromes. Using a PCR-based prototype assay (GeneXpert) capable of accurately measuring five mRNA transcripts directly from whole blood, the primary objective of this study is to determine the ability to differentiate infectious versus non-infectious febrile syndromes in patients as compared to the results of traditional pathogen-based microbiology tests in an adjudicated cohort. Additionally, we explored surface maker determination using flow cytometry to detect potential surface markers that predict the etiology of fever, we aim to evaluate the sensitivity and accuracy of such markers to understand febrile syndromes better. Finally, patient pathways from the Emergency Department (ED) arrival to the inpatient unit admission were collected to map out and determine critical time points where such host-based diagnostic may have the highest potential for improving clinical decision-making for febrile patients.

## 2. Materials and Methods

### 2.1. Study Design and Study Participants

The study included 100 febrile patients recruited from a single site at Massachusetts General Hospital (MGH), and 22 healthy controls enrolled through an on-campus primary care clinic. The study was conducted according to the guidelines of the Declaration of Helsinki, and approval was obtained from the MGH institutional review boards (IRB) protocol, approval number: 2021P0003374. Informed consent was obtained from all subjects participating in the study. The research investigators designed the study, collected the data, and performed the analysis. DH Diagnostics LLC provided unrestricted funding and the GeneXpert^®^ system required to complete the study but was not involved in data interpretation, analysis, or assembling the manuscript.

### 2.2. Patients

Potential subjects were detected daily through generated screening EPIC reports that detect patients admitted to the MGH ED with a documented fever. Patients were enrolled within 24 h after ED presentation, and blood samples were collected within 48 h after ED presentation. Patients were considered eligible for recruitment if they were greater than 18 years of age, had a documented fever >38 °C, and had an ongoing work-up to determine the etiology of the fever initiated. Patients were excluded from the study in the case of pregnancy. Subjects were enrolled from March 2022 till October 2022.

### 2.3. The Prototype Host Response Assay Protocol

A prototype host response assay that integrates sample preparation and RT-PCR to measure 6-mRNA genes on the GeneXpert system was developed. The prototype assay measures five mRNA targets (RUNX1, LY6E, IRF9, ITGAM, PSTPIP2) and a control (ABL1) chosen from the literature based on their combined ability to distinguish infectious from non-infectious illness in patients with fever [15,16,17,18,19,20,21]. Assay testing was performed in the Division of Infectious Diseases at Massachusetts General Hospital in Boston, MA. A total of 20 cc of venous blood was collected in ethylenediaminetetraacetic acid (EDTA) tubes from the enrolled subject. A total of 200 μL of blood was added to the Xpert Lysis Reagent. Next, 1 mL of blood/lysis mixture was added to the GeneXpert cartridge and loaded into the GeneXpert system for processing. The turnaround time for assay results is 35 min. (Figure 1).

### 2.4. Preparation of Human Neutrophils

Healthy blood donors consented under the MGH IRB protocol, approval number: 2018P001283. Whole blood was collected in EDTA-coated vacutainers (Beckton Dickinson, Franklin Lakes, NJ, USA) and subsequently centrifuged at 1500× *g* for 15 min. Buffy coat was collected, and neutrophil isolation was performed using the negative selection EasySep Direct Human Neutrophil Isolation Kit, according to the manufacturer’s instructions (STEMCELL Technologies, Seattle, WA, USA). Wright-Giemsa staining was performed after the isolation process to confirm neutrophil purity from the isolation kit. Flow cytometry was also used to verify a high neutrophil purity from the isolation procedure (≥94% neutrophil purity). Cell concentration and viability were measured by staining the cells with a 1:10 dilution of acridine orange/propidium iodide followed by automatic cell counting using the LUNA fl Dual Fluorescence Cell Counter (Logos Biosystems, Annandale, VA, USA) (≥99% live).

### 2.5. Flow Cytometry

Isolated neutrophils (50,000 cells) were stained in fluorescent activated cell sorting (FACS) buffer containing 2% heat-inactivated fetal bovine serum (FBS) (Life Technologies, Dun Laoghaire, Ireland) and 1 mM EDTA (Life Technologies) in phosphate buffer saline (PBS) without calcium and magnesium (Corning, Corning, NY, USA). Cells were stained with antibodies for 30 min at 4 °C in FACS buffer containing (BV421) anti-CD10 (1:200 dilution; clone HI10a; BioLegend, San Diego, CA, USA) and (BV786) anti-CXCR4 (1:100 dilution; clone12G5; BioLegend) and (AF647) anti-CD64 (1:100 dilution; clone 10.1; BioLegend) or (BV786) anti-CD63 (1:100 dilution; clone H5C6; BioLegend) and (APC) anti-CD66b (1:800 dilution; clone G10F5; BioLegend), or (BV421) anti-CD62L (1:200 dilution; clone DREG-56; BioLegend), and (PE/Dazzle) anti-CD32 (1:200 dilution; clone FUN-2; BioLegend). Data were acquired on a BD FACSCelesta (BD Biosciences, San Jose, CA, USA) with a BVR laser configuration (488 nm, 405 nm, 640 nm). Before recording data, gates were prepared so that 10,000 neutrophil events could be collected. FCS files were exported from BD FACSDiva Software (BD Biosciences) in a 3.0 format. FCS files were analyzed using FlowJo v.10 software (BD Biosciences).

### 2.6. Outcomes

The primary outcome of the study is to determine the efficiency of the GeneXpert prototype assay monitoring host-response gene signatures in correctly predicting the etiology of fever when compared to microbiology results. The expression of the genes stratified by the different groups measured by delta Ct (ABL1 Ct value—Target Ct value), the correlation of the five genes with each other, and their predictive performance of the etiology of fever will be evaluated. Microbiology results were defined as the results of blood and urine cultures and viral panels. Febrile patients were categorized among two groups based on their infection status, and healthy controls were categorized as a third group.

Febrile Positive (FP) group (*n* = 74): Febrile patients with confirmed positive microbiology results and clinically adjudicated febrile patients with suspected infections by clinical assessment despite the absence of positive microbiology results were categorized into one group of positive composite outcomes labeled FP. Clinical adjudication was performed by manual physician chart review.

Febrile Negative (FN) group (*n* = 26): Febrile patients with negative microbiology results and the absence of suspected infection by clinical assessment were categorized into a second group labeled FN.

Healthy Control (HC) group (*n* = 22): Healthy subjects presenting to the primary clinic for routine annual laboratory testing were categorized into a third group labeled HC (See Figure 2).

### 2.7. Statistical Analyses

Descriptive measures such as median, interquartile range, frequencies, and percentages were used to summarize the data. Exact binomial confidence intervals were used to estimate confidence intervals for sensitivities and specificities. Wilcoxon rank-sum and Fisher exact tests were used to compare continuous and categorical study variables, respectively, between febrile positive and negative patients. For comparing continuous outcomes among the three groups (healthy, febrile negative, and febrile positive patients), the Kruskal–Wallis test with Dunn’s post-hoc analysis was used. Spearman’s rank correlation was used to assess the strength and direction of association between the study variables. Univariate logistic regression models were utilized to assess the predictors of being a febrile positive. The least absolute shrinkage and selection operator (Lasso) logistic regression algorithm was performed to select the most predictive variables of febrile positive. We estimated the predictive accuracy of the selected variables in distinguishing class membership (febrile negative or positive) using several machine-learning algorithms, including linear discriminant analysis (LDA), quadratic discriminant analysis (QDA), k-nearest neighbor (KNN), support vector machine (SVM), classification tree (CART), AdaBoost (ADA), neural networks (NNET), random forest (RF), Gaussian process and logistic regression. The leave-one-out cross-validation procedure was used to estimate the performance of the classifier algorithms. We used the algorithm with the highest cross-validated area under the receiver operating curves to evaluate the performance of the selected variables as biomarkers of febrile-positive patients. Statistical analyses were performed using the R package version 4.2.1 and SAS software version 9.4 (SAS Institute, Cary, NC, USA). All *p* values were 2-sided and considered statistically significant if <0.05.

## 3. Results

### 3.1. Patients

The median time for running the assay for all study subjects after blood collection was 3 h, and the average time was 3.4 h. The baseline characteristics of the patients are shown in Table 1. A total of *n* = 122 subjects were enrolled in the study and divided into three groups. The first group was the FP group, which comprised 74 patients (60.7%). Among the FP group, 45 patients (60.8%) were males. The confirmed infections included 57 bacterial and 17 viral infections. The bacterial infections included 17 urinary tract infections, 24 bloodstream infections, three cellulitis infections, two cholangitis infections, two pelvic inflammatory disease infections, one hepatic abscess infection, seven community-acquired pneumonia infections, and one endocarditis infection. The urinary tract and bloodstream infections were caused by Gram-positive (43%) and Gram-negative pathogens (57%). On the other hand, the viral infections included eight SARS-CoV-2 (COVID-19) infections, three influenza A infections, three rhinovirus infections, one human metapneumovirus infection, one Epstein-Barr virus (EBV) infection, and one parainfluenza virus infection.

The median age was 63 (interquartile range, 44–73). Thirty percent (29.7%) of the patients had a body-mass index (the weight in kilograms divided by the square of the height in meters) >30. In total, 13.5% of the patients had known diabetes mellitus, and 10.9% had a history of previous lung disease (e.g., asthma, COPD). A total of 15 patients (20.3%) had a history of malignancy, 22 patients (29.7%) had an active malignancy, 17 patients (23%) had congestive heart failure, 20 patients (27%) had a history of recurrent infections, and two patients (2.7%) had liver disease/cirrhosis. A total of 48 patients (64.9%) did not require supplemental oxygenation at day 1, 23 patients (31.1%) were receiving supplemental oxygen at ≤6 L per minute, delivered by nasal cannula, to maintain an oxygen saturation >92%, and three patients (4%) were receiving high-flow oxygen. The median white blood cell (WBC) count for the FP group was 12.71 × 10^9^/L (interquartile range: 6.2–16.2 × 10^9^), and the median C-reactive protein (CRP) was 149.25 mg/L (interquartile range: 54.3–261.3 mg/L).

The second group was the FN group, which comprised 26 patients (21.3%). Among the FN group, 17 patients (65.4%) were males. The median age was 57.5 (interquartile range, 27–66). The number of patients with a body-mass index (the weight in kilograms divided by the square of the height in meters) >30 was 7 (26.9%). In total, 15.4% of the patients had known diabetes mellitus, and 3.8% had a history of previous lung disease (e.g., asthma, COPD). One patient (3.8%) had a history of malignancy, ten patients (38.5%) had an active malignancy, two patients (7.7%) had congestive heart failure, four patients (15.4%) had a history of recurrent infections, and two patients (7.7%) had liver disease/cirrhosis. A total of 23 patients (88.4%) did not require supplemental oxygenation on day 1, 2 patients (7.7%) were receiving supplemental oxygen at ≤6 L per minute, delivered by nasal cannula, to maintain an oxygen saturation >92%, and 1 patient (3.8%) required mechanical ventilation. The median white blood cell (WBC) count for the FN group was 9.37 × 10^9^/L (interquartile range: 4.72–12.56 × 10^9^), and the median C-reactive protein (CRP) was 96.15 mg/L (interquartile range: 47.55–148.1 mg/L).

The third group was the HC group, consisting of 22 patients (18%). Among the HC group, six patients (27.3%) were males. The median age was 57 (interquartile range, 43–68). The number of patients with a body-mass index (the weight in kilograms divided by the square of the height in meters) >30 was 8 (36.4%). In total, 18.2% of the patients had known diabetes mellitus, three patients (13.64%) had a history of malignancy, and one patient (4.5%) had an active malignancy. (See Table 1).

### 3.2. Network Structure Correlation of the Five Genes in the FP, FN, and HC Groups

To understand the network structure of the five genes, correlation plots between the gene signatures delta Ct in the three different groups were performed. The results showed that the network structure representing the correlation of the five genes for febrile patients with positive microbiology (FP) or with negative microbiology (FN) using the GeneXpert prototype assay was significantly more robust than the network structure of the five genes in the HC group. In the febrile patients’ (FP and FN) group, a strong correlation with a positive correlation coefficient is demonstrated between the five genes. The FP group’s five genes (ITGAM, IRF-9, LY6E, PSTPIP2, and RUNX1) were significantly correlated with each other (*p*-value < 0.001). Similarly, the FN group’s five genes (ITGAM, IRF-9, LY6E, PSTPIP2, and RUNX1) were also significantly correlated with each other (*p*-value < 0.001). No significant difference in the association among the five genes was noted between the two groups of febrile patients.

On the other hand, in the HC group, a weak network structure with low correlation coefficients was noted between the five genes. Except for LY6E and IRF-9, which showed a significant correlation with a *p*-value < 0.05, the other gene signatures do not demonstrate any significant correlation. (See Figure 3.)

### 3.3. Expression Profiles of the Genes among Febrile and Non-Febrile Cohorts

To further investigate how the etiology of fever changes the host immune response, we next compared the gene expression profiles of matched subjects from the three different patient groups. We compared the expression of the five genes across the three different groups. Our analysis revealed that patients with confirmed infections (FP group) form a distinct cluster with higher expression of the five genes, demonstrating the effect of bacterial and viral infections on modifying the host response in febrile subjects. The analysis identified higher expression of the five genes in the FP group as compared to the FN and HC groups. Two clusters of genes were noted in the FP group. ITGAM, RUNX1, and PSTPIP2 formed one cluster with high expression in subjects with confirmed bacterial infections, whereas LY6E and IRF-9 formed another cluster with high expression, mainly in patients with confirmed viral infections. Moreover, for patients in the HC group, the expression levels of the five genes are lower than in other groups and are detected at low levels, implicating the role of fever on the immune response. (See Figure 4).

### 3.4. Predictive Performance of the GeneXpert Assay

Figure 5 displays the odds of being febrile with positive microbiology. In addition to the five genes, several other covariates, including age, gender, disease severity, cancer status, durations of fever, presence of fever on blood collection day, and antibiotics administration duration, were considered. The higher expression of the five genes was associated with higher odds of being febrile-positive patients:

IRF-9 (OR = 1.750, 95% CI = 1.16 to 2.638), ITGAM (OR = 1.533, 95% CI = 1.047 to 2.244), PSTPIP2 (OR = 2.191, 95% CI = 1.293 to 3.711), and RUNX1 (OR = 1.974, 95% CI = 1.069 to 3.646) had a significant expression with positive infection status. The odds of being febrile-positive were associated with higher values of the LY6E gene, although it did not reach statistical significance (OR = 1.11, 95% CI: 0.91–1.35).

Considering the differences in antibiotic exposure duration from the ED admission to the time of blood collection, and the potential effects of longer antibiotic usage on altering the gene expression profiles, stratification by antibiotic duration was also performed. Two groups of subjects were evaluated: those receiving antibiotics for two or fewer days before collecting blood and running the assay, and those receiving antibiotics for more than two days before collecting blood and running the assay. The results demonstrate that subjects receiving antibiotics for two or fewer days before running the assay (OR = 6.682, 95% CI = 2.452 to 18.206) were more significantly associated with positivity in infection status. The difference in the distribution of the fives genes varied by the duration of antibiotics usage, as shown in the density plots displayed in Figure 6. As expected, the differences between gene expression of febrile-positive and febrile-negative subjects are more noticeable in the shorter antibiotic usage category. The other parameters did not demonstrate any statistically significant association with infection status. We used machine-learning algorithms to develop a classifier model to classify study participants into groups based on the significant variables presented in Figure 5. The classifier model assessing the performance of the five genes in subjects receiving antibiotics for less than two days correctly classified more than 80% of the participants into their respective groups, i.e., FP or FN groups. The classifier correctly classified 65% [95% CI: 0.54–0.75] of the FP subjects and 89% [95% CI: 0.71–0.96] of the FN subjects.

### 3.5. Predictive Performance of Surface Markers to Determine Infection Status

Using flow cytometry to detect specific markers that can predict infection in 20 febrile patients of our study subjects (13 FP and 7 FN) and 9 HC subjects, three different surface markers, CD10 (percent positive), CD64 mean fluorescence intensity (MFI), and CXCR4 MFI demonstrated a significant distinction between the FP and FN groups. The expression of CD10 and CXCR4 were significantly lower in the FP group compared to the FN group. In contrast, the expression of CD64 was significantly higher in the FP group compared to the FN group (See Figure 7).

### 3.6. Patient Care Pathways

Different pathways mapping the clinical course of patients (*n* = 70) from the point of the initial evaluation in the ED arrival until the inpatient unit admission were collected to determine critical time points in clinical decision-making. Such pathways provided a framework where the implementation of the GeneXpert prototype assay could be best utilized. Average times to first labs, imaging, cultures, and antibiotics were collected for FP patients with bacterial infections (*n* = 44), FP patients with viral infections (*n* = 12), and FN patients (*n* = 14). The following timepoints were collected for the enrolled subjects prior to the establishment of a definite diagnosis. All patients were being managed for possible infections based on their febrile illness. The results show that the average times to the first labs are 1.09 h, 1.3 h, and 2 h for patients with bacterial, viral, and no infections, respectively. The average times for the first cultures performed were 2 h, 2.1 h, and 2.8 h for patients with bacterial infections, viral infections, and no infections, respectively. The average times to first imaging performed were 2.4 h, 2.6 h, and 3.5 h for patients with bacterial infections, viral infections, and no infections, respectively. Finally, the average times to first antibiotics administration were 4.9 h, 4.2 h, and 3.9 h for patients with bacterial infections, viral infections, and no infections, respectively. There were no significant differences between any of the average times among the three different groups indicating the need for earlier diagnostic to guide better management of febrile etiologies. It is noteworthy to mention that out of twelve patients with viral infections, eight patients (67%) received antibiotics in the ED prior to viral diagnostics indicating a viral pathogen. Four out of these eight (50%) were continued on antibiotics after the viral diagnostics turned positive due to concern for bacterial superinfection. Out of 14 patients with no infections (FN), 7 patients (50%) received antibiotics in the ED before ruling out infectious etiologies, and out of 44 patients with bacterial infections, 43 patients (97.7%) received antibiotics in the ED. (See Figure 8a).

Moreover, the average time from ED arrival to inpatient admission was collected for these patients. The average times to admission were 21.1 h, 18.6 h, and 23.4 h for patients with bacterial infections, viral infections, and no infections, respectively. (See Figure 8b).

## 4. Discussion

In this prospective study, a robust predictive performance in differentiating the etiology of febrile syndromes has been demonstrated by the GeneXpert prototype assay. Such results reveal promise for using this assay for accurate diagnostic outcomes in the future. The rapid turnaround time of the assay (35 min) and the simple sample preparation protocol could provide additional potential for future implementation of such a modality from both clinical and possibly economic perspectives.

Notably, the role of each of the five genes (ITGAM, IRF-9, LY6E, PSTPIP2, and RUNX1) has been linked to response to infection in the literature. To start with, ITGAM is known to promote the adherence of monocytes and macrophages and to mediate the uptake of opsonized particles and pathogens [22]. In fact, in one recent study, using a four-gene signature that includes ITGAM and three other genes has demonstrated a promising model to diagnose patients with sepsis [23]. Additionally, IRF-9 has also been shown to be an integral transcription factor in mediating the type I interferon antiviral response, and the expression of IRF-9 plays an essential role in antiviral immunity [24]. In a recent case report, a five-year-old child with IRF-9 deficiency experienced severe influenza pneumonitis, further highlighting IRF-9′s role in antiviral immunity [25]. Similarly, Ly6E genes have also been shown to possess an antiviral regulation response. Ly6E confers critical antiviral functions by restricting the entry of human coronaviruses, including SARS-CoV, MERS-CoV, and SARS-CoV-2, by interfering with spike protein-mediated membrane fusion [26].

Interestingly, our results revealed two gene expression clusters in patients with confirmed infections. ITGAM, RUNX1, and PSTPIP2 formed one cluster with high expression in subjects with confirmed bacterial infections, whereas LY6E and IRF-9 formed another cluster with high expression in patients with confirmed viral infections. These findings further support the different roles of the host-response gene signatures regarding the etiology of infections.

In this study, flow cytometry was performed to evaluate the role of host surface markers in response to infections. The results showed that three different surface markers, CD10, CD64, and CXCR4, significantly differentiate between the presence and absence of infection in patients with febrile illnesses. In a previous study, the role of CD10 expression in sepsis patients was described, where CD10 and CD66b were shown to be effective biomarkers and good predictors for early bacterial infections in patients with suspected sepsis. When compared to the performance of procalcitonin and CRP, the accuracy of CD10 and CD66b expression for predicting bacterial infections was significantly higher, with a sensitivity of 86.5% and a specificity of 90.3%. [27]. Other studies have described the role of CD64 in differentiating bacterial and viral infections [28]. The immunologic assays performed in this study were exploratory. Consequently, future considerations directed at understanding the role of cell surface markers in the setting of infections can provide insight into developing a combined model that detects gene signatures and surface markers for more accurate diagnostic performance. Larger sample sizes will be ultimately needed to run such a classifier model.

Thus, from a clinical perspective, the development of this promising five-gene signature assay has the potential to serve as a guide for better patient outcomes. The rapid and accurate differentiation between infectious and non-infectious etiologies in patients presenting with fever can lead to more optimal administration of antibiotics and a reduction in antimicrobial resistance [29,30]. Additionally, the results from this study suggest a potential effect of longer antibiotic administration duration on the expression profiles of the five genes. The prototype assay demonstrated more accurate performance for febrile subjects receiving shorter antibiotic duration before running the assay [31], attributed to possible gene expression changes in response to exposure to antimicrobials.

Moreover, patient pathways collected in this study show that 50% of patients with no infections and 67% of patients with viral infections receive unnecessary antibiotics in the ED due to the absence of a rapid modality that can differentiate the etiology of fever [32]. The implications of our findings suggest that the optimal implementation of such a diagnostic would be in the ED setting before the administration of antibiotic therapy. Such an implementation has the potential to improve patient outcomes and provide optimal antibiotic use. The reduction in antibiotic usage, in addition to reducing antimicrobial resistance, ultimately leads to a reduction in antibiotics-associated gastrointestinal, dermatologic, musculoskeletal, hematologic, renal, cardiac, and neurologic adverse events [33].

Finally, from an economic perspective, such a modality could provide valuable decision-making guidance regarding unnecessary hospital admissions for the management of infections for patients with low suspicion of infectious etiologies. In a recent study of inpatient hospital costs for COVID-19 patients in the United States, the overall median cost of a hospital stay per day was shown to be more than $1700 USD, while the overall median cost of an ICU admission per day was shown to be approximately $3000 USD [34]. Thus, the reduction in patients’ hospital length of stay duration could potentially lead to significant healthcare savings [35]. Additional reductions in healthcare costs associated with adopting such an assay include a decline in the number of blood cultures drawn and fewer antibiotics being prescribed, decrease in unnecessary laboratory tests, imaging, and procedures [36,37]. Future interventional trials will be required for validation. In short, attempts have been focused on using the host response as a reference for a more personalized approach to precision medicine, taking into account its impact. By understanding the mechanisms underlying the host-immune response using specific signatures and robust diagnostics, we can achieve better medical decision-making guidance that ultimately leads to better patient outcomes.

### Limitations

Our study should be interpreted in the setting of several limitations. One significant limitation is regarding the classification of subjects into different groups. Patients with suspected, and often apparent, infections such as cellulitis or community-acquired pneumonia had no cultures taken and no confirmed positive results. Thus, for the accurate classification of these patients, the clinical assessment of the medical team was followed. Moreover, the variability in the duration of antibiotic therapy received by the patients before the blood sample collection may present a confounding variable that alters the host’s immune response. Additionally, the GeneXpert prototype demonstrated lower predictive performance for patients on longer antibiotic duration suggesting a limited optimal timing for implementation of this diagnostic. Another limitation of the study is that the GeneXpert is designed to detect the expression of genes that point toward bacterial and viral infections. Its performance in detecting fungal and parasitic infections is yet to be evaluated. The lack of an adequate sample size to perform subgroup analysis between bacterial and viral infections poses an additional limitation. Larger cohorts will be required to perform any subgroup analysis for differentiating the microbiological etiology. Finally, the results presented in this manuscript are based on a single study center. A multi-center study might be essential to attain a larger and more diverse sample size.

## 5. Conclusions

The implementation of novel rapid host-based diagnostics, such as the GeneXpert prototype assay and flow cytometry for patients with febrile syndromes has the potential to reduce adverse events, decrease the misuse of antibiotics, and lower the rate of emerging antimicrobial resistance. Furthermore, such modalities have the potential to reduce healthcare costs and inform clinicians about the optimal utilization of resources from an economic perspective. Future directions could be directed toward launching an interventional trial to assess the efficacy of the assay in reducing healthcare costs and patient adverse outcomes. Additionally, enrolling a larger sample size for subgroup analysis to assess the performance of the GeneXpert in differentiating bacterial vs. viral infections will be necessary. Finally, the future development of an exploratory model using the multi-modality host-based assays including flow markers and best-performing gene signatures may provide improved diagnostic accuracy in the management of the undifferentiated febrile patient.

## Figures and Tables

**Figure 1 diagnostics-13-00953-f001:**
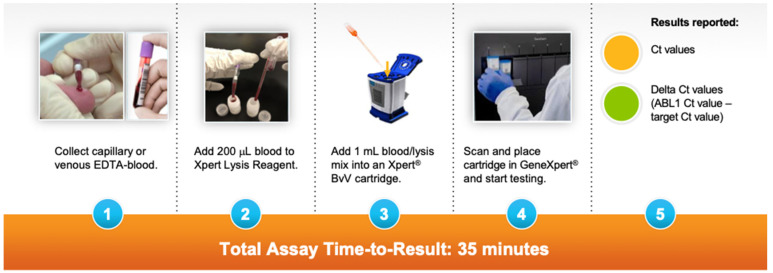
Steps for running the prototype GeneXpert assay.

**Figure 2 diagnostics-13-00953-f002:**
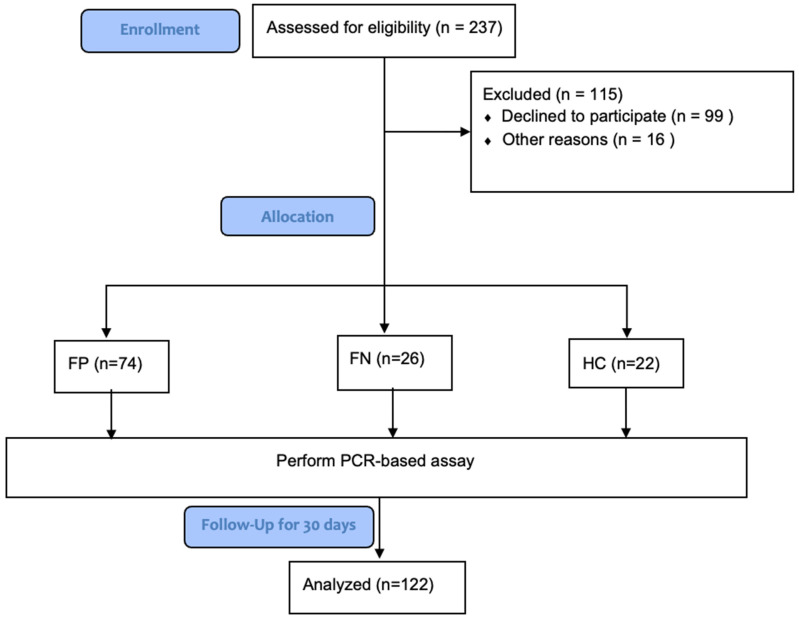
Study Flowchart.

**Figure 3 diagnostics-13-00953-f003:**
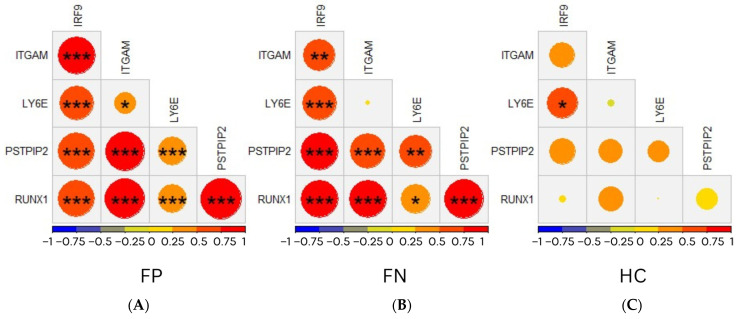
Correlation plots between the five genes in (**A**) FP group, (**B**) FN group, and (**C**) HC group. The size of the circle indicates the strength of the correlation, with the larger circle representing a stronger correlation. The value inside the circle represents the significance level (*** < 0.001; ** < 0.01, * < 0.05).

**Figure 4 diagnostics-13-00953-f004:**
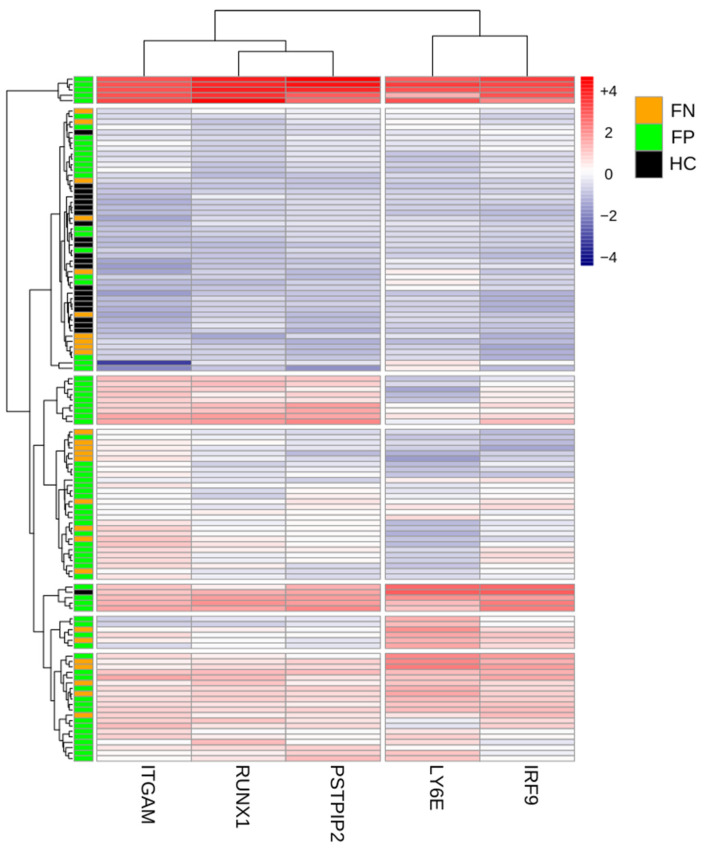
Heat map of the five expressed genes from the FP group, FN group, and HC group. Rows represent the subjects, and columns represent genes. Gene expression levels are color-coded; red and blue represent high (+4) and low (−4) expression levels, respectively.

**Figure 5 diagnostics-13-00953-f005:**
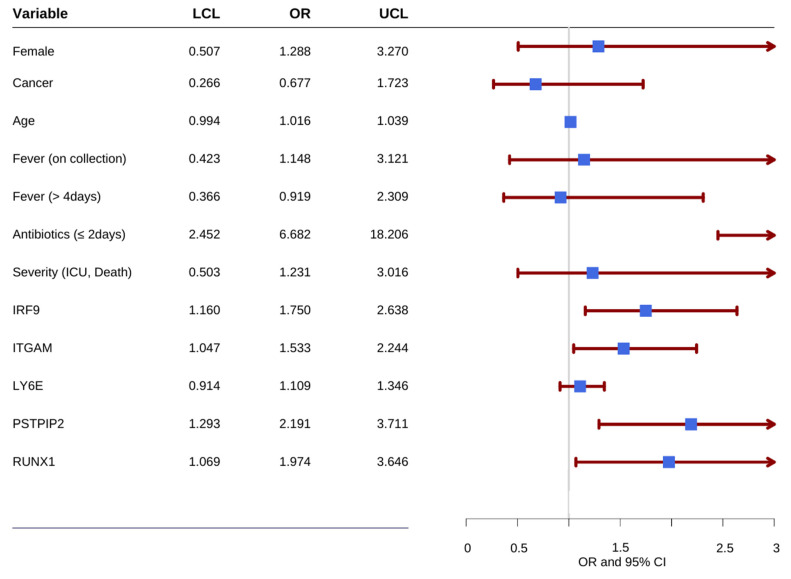
Forest plot presenting odds of FP.

**Figure 6 diagnostics-13-00953-f006:**
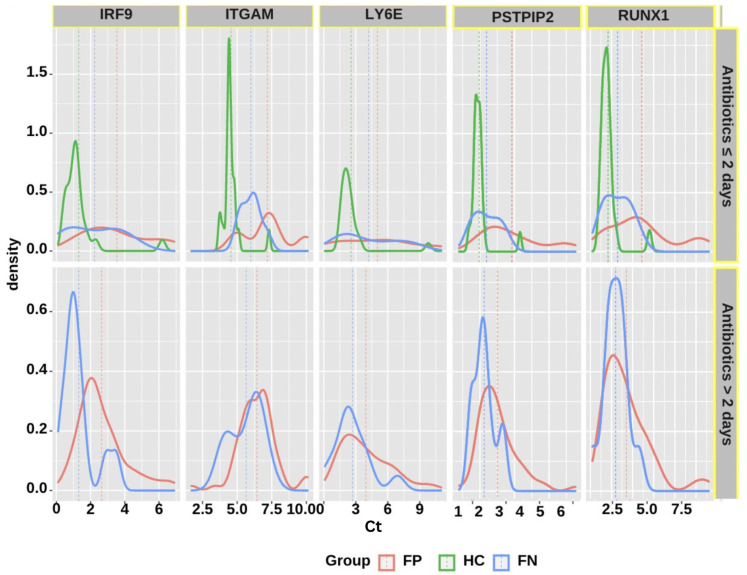
Density plots displaying distributions of the five genes in the FP, FN, and HC groups classified by antibiotic duration status.

**Figure 7 diagnostics-13-00953-f007:**
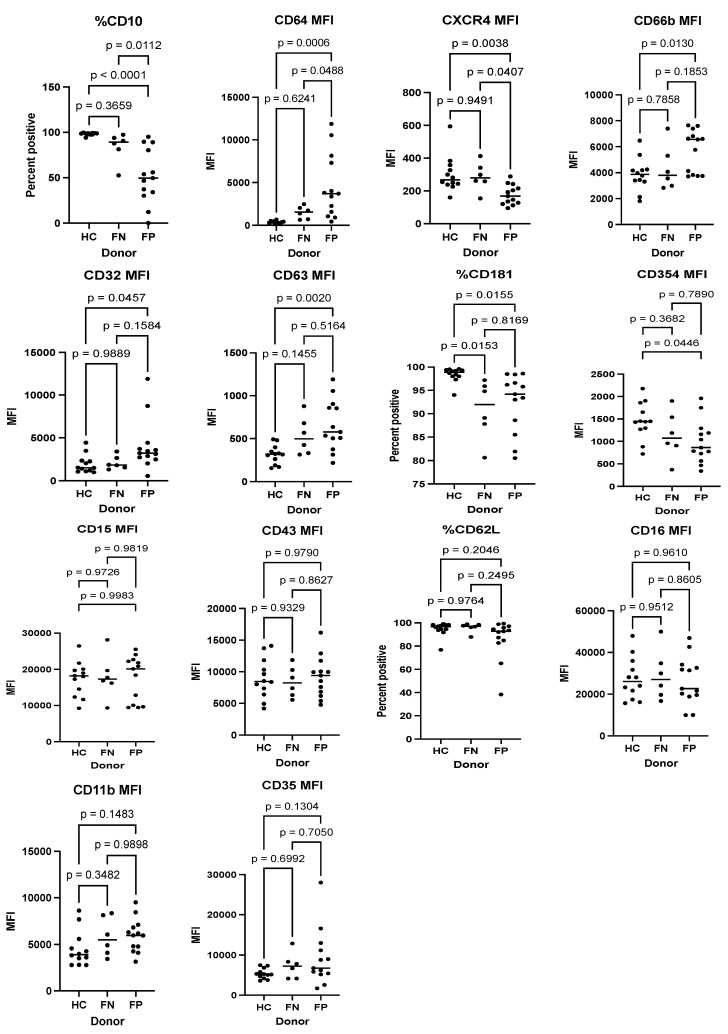
Distribution of surface markers using flow cytometry for FP, FN, and HC group.

**Figure 8 diagnostics-13-00953-f008:**
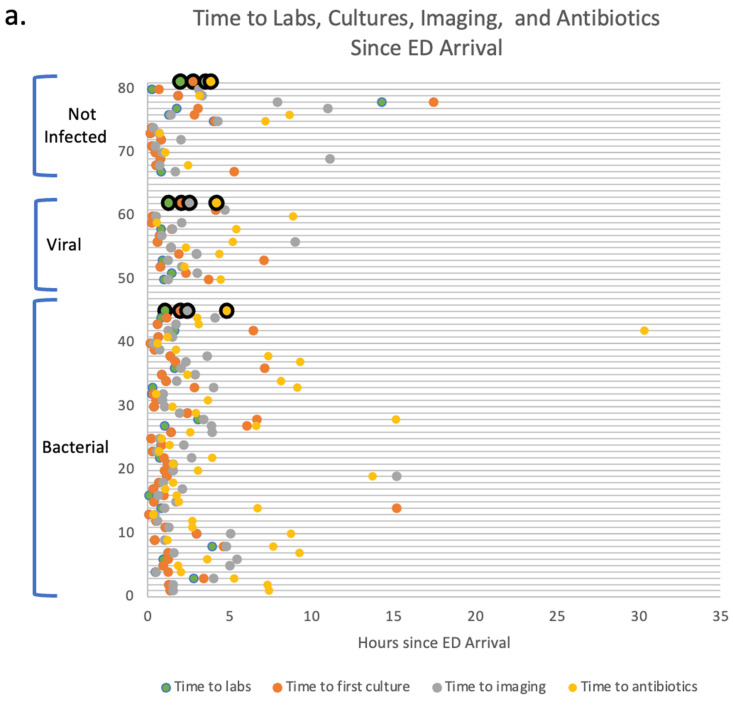
(**a**) Primary care pathways for the FP, and FN groups from ED arrival to admission represent the time to first labs, first cultures, first imaging, and first antibiotics. Large circles with black borders represent average hours. Small circles represent the timepoints for each subject for the FP and FN groups. (**b**) Average time from ED arrival to admission for the FP and FN groups. Red circles represent average hours for time to admission. Blue circles represent the time from ED arrival to admission for each subject for the FP and FN groups.

**Table 1 diagnostics-13-00953-t001:** Patient demographics and characteristics.

Characteristic	Febrile Positive	Febrile Negative	Healthy Controls	Total
	No. (%)	No. (%)	No. (%)	No. (%)
	(*n* = 74)	(*n* = 26)	(*n* = 22)	(*n* = 122)
Demographics				
Age, Median (IQR)	63 (44–73)	57.5 (27–66)	57 (43–68)	60.5 (41–70)
Men	45 (60.8)	17 (65.4)	6 (27.3)	68 (55.7)
BMI > 30	22 (29.7)	7 (26.9)	8 (36.4)	37 (30.3)
Race				
Non-Hispanic	72 (97.3)	26 (100)	22 (100)	120 (98.36)
Hispanic	2 (2.7)	0 (0)	0 (0)	2 (1.64)
Oxygen Requirement at day 1				
No requirement	48 (64.9)	23 (88.4)	22 (100)	93 (76.2)
Nasal Cannula	23 (31.1)	2 (7.7)	0 (0)	25 (20.5)
High Flow Nasal Cannula	3 (4)	0 (0)	0 (0)	3 (2.5)
Intubation	0 (0)	1 (3.8)	0 (0)	1 (0.8)
Medical History				
Diabetes	10 (13.5)	4 (15.4)	4 (18.2)	18 (14.8)
COPD	8 (10.9)	1 (3.8)	0 (0)	9 (7.4)
History of Malignancy	15 (20.3)	1 (3.8)	3 (13.64)	19 (15.6)
Hypertension	25 (33.8)	7 (26.9)	5 (22.7)	37 (30.3)
Organ Transplant	7 (9.5)	0 (0)	0 (0)	7 (5.7)
Coronary artery disease	7 (9.5)	0 (0)	0 (0)	7 (5.7)
Heart Failure	17 (23)	2 (7.7)	0 (0)	19 (15.6)
Active Malignancy	22 (29.7)	10 (38.5)	1 (4.5)	33 (27)
CKD	9 (12.2)	0 (0)	1 (4.5)	10 (8.2)
Liver Disease/Cirrhosis	2 (2.7)	2 (7.7)	0 (0)	4 (3.3)
Recurrent Infections	20 (27)	4 (15.4)	0 (0)	24 (19.7)
Substance use disorder	2 (2.7)	2 (7.7)	0 (0)	4 (3.3)
Neurological Disease	9 (12.2)	2 (7.7)	0 (0)	11 (9)
Median Laboratory Values (IQR)				
WBC Count – × 10^9^/L	12.71 (6.2–16.2)	9.37 (4.72–12.56)	N/A	11.45 (5.76−15.78)
C-reactive protein	149.25 (54.3–261.4)	96.15 (47.55–148.1)	N/A	136.05 (54.3−242.7)

## Data Availability

The data underlying this article are available in the article.

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
