# Peer review of "Novel Host Response-Based Diagnostics to Differentiate the Etiology of Fever in Patients Presenting to the Emergency Department [Author-notes fn1-diagnostics-13-00953]"

_diagnostics, 2023, doi:10.3390/diagnostics13050953_

Round 1
Reviewer 1 Report
Atallah et al., report a novel diagnostic gene set to stratify individuals presenting at the emergency department based on the etiology of their fever. They had three group, a control group, a febrile positive and febrile negative group. Two genes LY6E and IRF-9 formed a distinct cluster that were associated with viral infections and three genes, ITGAM, RUNX1 and PSTPIP2 formed a separate cluster associated with bacterial infections. The authors also showed 3 immune signatures, CD10, CD64 and CXCR4CFMI showed differences between FP and FN groups. The authors showed that their signature was very effective in aiding diagnosis of individuals with less than 2 days of antibiotic usage. This work is commendable.
Specific comments.
The quality of diagrams is not really good as it is fuzzy and Fig 4 genes, numbers is hard to decipher. Fig 8b red dots do not seem to align for the different groups. The legends do not sufficiently describe abbreviations and significance levels.
The references provided for in the introduction are old. Reported false negativity rates, high labour (line 60) must be reviewed in light of more recent literature and more recent diagnostic tools.
Line 113 lack a citation.
Line 391-392 needs to be discussed based on the study results. In the present form, literature is just cited.
Line 403, I think the authors should therefore express this as a limitation of the diagnostic for individuals undergoing antibiotic treatment based on their findings. The statement appears incomplete.
The sensitivity of combining geneXpert and immunology can be included to show how effective the diagnostic is.
Author Response
Response to Reviewer 1 Comments
Point 1: The quality of diagrams is not really good as it is fuzzy and Fig 4 genes, numbers is hard to decipher. Fig 8b red dots do not seem to align for the different groups. The legends do not sufficiently describe abbreviations and significance levels.
Response 1: Thank you for this comment.
For figure 4, I have fixed the resolution and removed the numbers representing the patient IDs. I have also added a clearer legend.
For Figure 8a and b, I have fixed the red dots to align correctly for the different groups and have added additional descriptions to the legends.
Please see Figure 4 and Figure 8.
Point 2: The references provided for in the introduction are old. Reported false negativity rates, high labour (line 60) must be reviewed in light of more recent literature and more recent diagnostic tools.
Response 2: Thank you for this comment.
I have reviewed and changed the limitations of pathogen-based testing, specifically blood cultures, to low sensitivity and low specificity limitations. I have also replaced the references provided with more recent literature taking into account more recent diagnostic tools.
Please see track changes in the introduction section.
Point 3: Line 113 lack a citation.
Response 3: Thank you for this comment.
I have added the seven reference studies from the literature that guided the decision to choose the five mRNA targets (RUNX1, LY6E, IRF9, ITGAM, PSTPIP2) for developing the GeneXpert prototype assay.
Please see track changes in the methods and references section.
Point 4: Line 391-392 needs to be discussed based on the study results. In the present form, literature is just cited.
Response 4: Thank you for this comment.
I have now extensively described the results of the study. I have reported the sensitivity and specificity of using CD10 for predicting bacterial infections in patients with sepsis and have described its accuracy when compared to the use of procalcitonin and CRP biomarkers.
Please see the track changes in the discussion section.
Line 403, I think the authors should therefore express this as a limitation of the diagnostic for individuals undergoing antibiotic treatment based on their findings. The statement appears incomplete.
Thank you for this comment.
I have added the lower performance of the GeneXpert for patients on longer antibiotic therapy as a limitation. Please see track changes in the limitation section.
Possible explanations could be that patients on longer antibiotic therapy before running the assay are more likely to be improving from their febrile illness. This could explain the lower performance of the diagnostic at that time point, the same way cultures cease to turn positive after longer antibiotic durations.
Point 5: The sensitivity of combining geneXpert and immunology can be included to show how effective the diagnostic is.
Response 5: Thank you for this important comment.
We do not have the power to run the classifier model for combining the GeneXpert and immunology results due to the difference in the sample size between the two assays.
Please note that 100 febrile subjects were tested using the GeneXpert assay, but only 20 febrile subjects were tested for surface markers. If we combine the two datasets, we will not have an adequate number to run a classifier model. Immunologic assays performed in this study were exploratory, and future studies with larger sample sizes will be needed.
Reviewer 2 Report
Johnny Atallah and colleagues report on a new method that can be used to differentiate the etiology of fever in patients presenting to the emergency department. In particular, the paper focus on differentiating infectious vs non-infectious febrile syndromes. The study included 100 febrile patients and 22 health controls. Using a novel PCR-based assay measuring 5 mRNA transcripts, the authors were able to differentiate infectious versus non-infectious febrile syndromes. Using advanced statistical modelling, the study was to correctly classify >80% of patients in their respective groups. This is an important topic as a timely identification of non-infectious febrile syndromes can reduce the use of antibiotics. I have several comments.
Comments
1. The study reports that >80% of individuals were correctly classified. It is, however, unknown how many individuals with infectious febrile syndrome were classified as having non-infectious febrile syndrome. This is important to know as such incorrect classifications can have an important impact as patients could not receive (effective) antibiotic treatment in a timely manner. The authors should therefore add the sensitivity of the newly developed method to the paper.
2. Similarly, it is also important to know how many patients having non-infectious febrile syndrome are correctly classified. Please add the specificity to the paper.
3. Please adapt the flow chart (Figure 2). It now seems that only samples from febrile negative patients were analyzed using PCR testing.
4. Please add p-values to Figure 7. (reporting p values as ns is not sufficient).
5. Line 339-340: “Average times for the first cultures performed were 2 hours, 2.1 hours, and 2.8 hours for 340 patients with bacterial infections, viral infections, and no infections, respectively.” Were cultures performed to diagnose viral infections? This is not the way that viral infections are normally diagnosed.
6. Line 343-344: “Finally, the average times to first antibiotics administration were 4.9 hours, 4.2 hours, and 3.9 hours for patients with bacterial infections, viral infections, and no infections, respectively” Please rephrase as I do not believe that antibiotics were given for viral infections.
7. Was study performed during COVID-19? A larger number of people having a viral infection could in such a period have been admitted than in other periods.
8. What diagnosis was made in FP patients? This could have an impact on the outcome.
Author Response
Response to Reviewer 2 Comments
Point 1: The study reports that >80% of individuals were correctly classified. It is, however, unknown how many individuals with infectious febrile syndrome were classified as having non-infectious febrile syndrome. This is important to know as such incorrect classifications can have an important impact as patients could not receive (effective) antibiotic treatment in a timely manner. The authors should therefore add the sensitivity of the newly developed method to the paper.
Response 1: Thank you for this comment.
We agree that this is important to know because it can impact the medical treatment that patients receive. Of 74 subjects with infectious febrile syndromes, 48 subjects (65%) were correctly classified.
Please see the track changes in section 3.4.
Point 2: Similarly, it is also important to know how many patients having non-infectious febrile syndrome are correctly classified. Please add the specificity to the paper.
Response 2: Thank you for this comment. Of 26 subjects with non-infectious febrile syndromes, 23 subjects (89%) were correctly classified.
Please see the track changes in section 3.4.
Point 3: Please adapt the flow chart (Figure 2). It now seems that only samples from febrile negative patients were analyzed using PCR testing.
Response 3: Thank you for this comment.
The flow chart has been fixed to show that the three different groups were analyzed using PCT testing.
Please see Figure 2.
Point 4: Please add p-values to Figure 7. (reporting p values as ns is not sufficient).
Response 4: Thank you for this comment.
P-values were added to Figure 7.
Please see figure 7.
Point 5: Line 339-340: “Average times for the first cultures performed were 2 hours, 2.1 hours, and 2.8 hours for 340 patients with bacterial infections, viral infections, and no infections, respectively.” Were cultures performed to diagnose viral infections? This is not the way that viral infections are normally diagnosed.
Response 5: Thank you for this comment.
All the time points in the ED were collected before establishing a definite diagnosis for the febrile patients. Cultures were performed to diagnose undifferentiated febrile patients before viral panels’ results indicated a positive infection.
The following statement was rephrased to explain why cultures were collected and to note that there was no significant difference in the management of patients whose blood cultures indicated a bacterial infection later on, patients whose viral panels indicated a viral infection later on, and patients who did not end up having any infection.
Please see track changes in section 3.6
Point 6: Line 343-344: “Finally, the average times to first antibiotics administration were 4.9 hours, 4.2 hours, and 3.9 hours for patients with bacterial infections, viral infections, and no infections, respectively” Please rephrase as I do not believe that antibiotics were given for viral infections.
Response 6: Thank you for this comment.
All the timepoints in the ED were collected before the establishment of a definite diagnosis for the febrile patients. Antibiotics were administered to undifferentiated febrile patients before viral panels results indicated a positive infection.
The following statement was rephrased to explain when the antibiotics were administered and to note that 67% of patients whose viral panels indicated a viral infection later on, and 50% of patients who did not end up having any infection, received antibiotics in the ED before the definite diagnosis was made.
Please see track changes in section 3.6
Point 7: Was study performed during COVID-19? A larger number of people having a viral infection could in such a period have been admitted than in other periods.
Response 7: Thank you for this comment.
The enrollment of subjects for this study was performed from March 2022 to October 2022. Eight patients (6.5% of the total sample size) were diagnosed with COVID-19.
Point 8: What diagnosis was made in FP patients? This could have an impact on the outcome.
Response 8: Thank you for this comment.
57 subjects had bacterial infections and 17 subjects had viral infections.
A detailed breakdown of the FP patients’ infections is now added to section 2.6.
Reviewer 3 Report
Because Fever is one of the among the most common presentations to urgent-care services and is linked to multiple disease processes the authors described the method which can help to develop and rapidly determine the etiology of fever in patients presenting to the ED, improved diagnostic modalities. In conducted studies authors presented/indicated the obtained results from 100 hospitalized febrile positive (FP) and febrile negative (FN) patients in terms of infection, and 22 healthy controls (HC). Authors evaluated the utility of a novel method based on PCR-based assay measuring five mRNA transcripts isolated directly from whole blood of examined human patients 1) ITGAM,
2) IRF-9,
3) LY6E,
4) PSTPIP2,
5) RUNX1)
These five mRNA transcripts were chosen to differentiate infectious versus non-infectious febrile syndromes as compared to traditional old and time consuming archaic methods pathogen-based microbiology results. The FP and FN groups observed a robust network structure with a significant correlation between the five genes (p-value<0.001). What was more important and significant the authors described and indicated that there was a statistically significant association between positive infection status and the following of the four genes: IRF-9 (OR=1.750, 95% CI=1.16-2.638), ITGAM (OR=1.533, 95% CI=1.047-2.244), PSTPIP2 (OR=2.191, 95% CI=1.293- 3.711), and RUNX1 (OR=1.974, 95% CI=1.069-3.646). Authors have been developed a classifier model to classify study participants based on these five genes and other demographic and clinical variables of interest to assess the discriminatory power of the genes. The model which have been described by the authors is the classifier model correctly classified more than 80% of the participants into their respective groups, i.e., FP or FN. This GeneXpert prototype holds promise for implementation as a modality for rapid clinical decisions that could reduce healthcare costs and improve outcomes in undifferentiated febrile patients presenting for urgent evaluation.
The manuscript is very interesting from an epidemiological point of view, and its content is a very important link that should be taken into account during routine work in emergency departments where sometimes only time decides about people's lives. The work is written correctly and understandably. It has appropriate sections and the results have been presented in an understandable way without raising any doubts.
I rate the work very highly and believe that it deserves publication in the MDPI journal
Author Response
Dear Reviewer 3,
We would like to thank you for your comments on the manuscript.
We appreciate your time, effort, and the positive feedback you provided.